# Sex differences in diabetic foot ulcer severity and outcome in Belgium

An-Sofie Vanherwegen[1]*, Patrick Lauwers[2], Astrid Lavens[1], Kris Doggen[1], Eveline Dirinck[3], on behalf of the Initiative for Quality Improvement and Epidemiology in multidisciplinary Diabetic Foot Clinics (IQED-Foot) Study Group[¶]

1 Health Services Research, Sciensano, Brussels, Belgium, 2 Thoracic and Vascular Surgery, University Hospital Antwerp, Edegem, Belgium, 3 Endocrinology, Diabetology and Metabolism, University Hospital Antwerp, Edegem, Belgium

¶ Membership of the Initiative for Quality Improvement and Epidemiology in multidisciplinary Diabetic Foot Clinics (IQED-Foot) Study Group is provided in the Acknowledgments.
* an-sofie.vanherwegen@sciensano.be

## Abstract

### Background

Sex differences are increasingly recognized to play an important role in the epidemiology, treatment and outcomes of many diseases. This study aims to describe differences between sexes in patient characteristics, ulcer severity and outcome after 6 months in individuals with a diabetic foot ulcer (DFU).

### Methods

A total of 1,771 patients with moderate to severe DFU participated in a national prospective, multicenter cohort study. Data were collected on demographics, medical history, current DFU and outcome. For data analysis, a Generalized Estimating Equation model and an adjusted Cox proportional hazards regression were used.

### Results

The vast majority of patients included were male (72%). Ulcers in men were deeper, more frequently displaying probe to bone, and more frequently deeply infected. Twice as many men presented with systemic infection as women. Men demonstrated a higher prevalence of previous lower limb revascularization, while women presented more frequently with renal insufficiency. Smoking was more common in men than in women. No differences in presentation delay were observed. In the Cox regression analysis, women had a 26% higher chance of healing without major amputation as a first event (hazard ratio 1.258 (95% confidence interval 1.048–1.509)).

### Conclusions

Men presented with more severe DFU than women, although no increase in presentation delay was observed. Moreover, female sex was significantly associated with a higher

**Data Availability Statement:** Data cannot be shared publicly because of the use of

pseudonomyzed patient data. Actors wanting to access (parts of the) data require an approval from the Belgian Information Security Committee Social Security and Health. For more information about the access procedure: iqedfoot@sciensano.be. Metadata (e.g. overview of variables, legal framework) are available on https://fair.healthdata.be/.

**Funding:** The IQED-Foot quality initiative is funded by the National Institute of Health and Disability Insurance. The funders had no role in study design, data collection and analysis, decision to publish, or preparation of the manuscript.

**Competing interests:** The authors have declared that no competing interests exist.

probability of ulcer healing as a first event. Among many possible contributing factors, a worse vascular state associated with a higher rate of (previous) smoking in men stands out.

## Background

Diabetic foot ulcers (DFU) are a common, yet devastating complication of longstanding diabetes that is strongly associated with peripheral arterial disease, peripheral neuropathy and foot deformity. The annual incidence of a DFU in people with diabetes ranges from 0.2% to 11%, depending on the clinical setting [1]. It is estimated that 19 to 34% of people with diabetes will develop a DFU in their lifetime. In addition, the recurrence rates are very high [2]. DFUs are associated with significant morbidity and a higher risk of lower limb amputation [3]. DFU and amputations have a major impact on the patient's quality of life [4,5] and on the burden on and cost to the health care system [1,6]. In this regard, early detection of the development of new lesions and close follow-up of existing lesions are crucial to improve outcomes in DFU patients.

Sex differences are increasingly recognized to play an important role in many aspects of health, such as epidemiology, pathophysiology, disease perception, treatment and outcomes [7]. Male sex has been identified as a risk factor for the development of DFUs [8]. Moreover, sex can strongly influence foot care behaviors [9,10]. There are differences in how men and women manage their diabetes and adhere to the care needed to prevent complications such as DFU [9–12].

Despite the acknowledgement of a negative role of male sex in the onset of DFU [8], the literature is less extensive on the differences in clinical presentation and outcomes between men and women. However, a better understanding will contribute to the optimization of care for this diabetes complication in a sex-specific manner. Therefore, the aim of this study was to identify sex differences in co-morbidities, referral pattern, ulcer severity at presentation, and outcome during a 6-month prospective cohort study in patients with DFU treated in Belgian multidisciplinary diabetic foot clinics (DFC).

## Methods

### Data collection

In 2005, a national diabetic foot care program was established in Belgium. The Initiative for Quality Improvement and Epidemiology in multidisciplinary Diabetic foot clinics (IQED--Foot) monitors the foot care provided in multidisciplinary DFCs. Every two years, each DFC prospectively collects data of the first 52 individuals presenting with a moderate to severe DFU or an active Charcot foot during the inclusion year and the evolution of the DFU over a period of 6 months. IQED-Foot is thus set up as a prospective cohort study. The national, aggregated data are analyzed by Sciensano, the Belgian health institute, and results are published in a public report. In addition, each DFC receives individual feedback on the care provided and quality improvement is encouraged by anonymous benchmarking [13].

Data were used from the 2018–2019 IQED-Foot data collection. During the inclusion period between January 1st, 2018 and December 31st, 2018, 35 recognized DFCs prospectively included a minimum of 52 patients who met the inclusion criteria. After inclusion, the patients were followed for 6 months. Sciensano has permission from the Social Security and Health chamber of the Belgian Information Security Committee to collect and use patient data within the IQED-Foot database. The processing of personal data was permitted under the legal basis

of general interest (Article 6(1)(e) and Article 9(2)(j) General Data Protection Regulation (GDPR)), and therefore did not require an informed consent. All data were pseudonymized by a trusted third party.

### Inclusion criteria

To be included in the study population, individuals had to be 18 years old or older, have diabetes mellitus (type 1, type 2 or other) and present in the DFC with a new DFU of Wagner grade 2 or higher [14], with or without an active Charcot on the same foot, during the inclusion period. In case the patient presented with multiple DFU, only the DFU with the highest expected impact on prognosis was included as the index DFU. Duplicate patients were identified across the 35 DFCs and only the episode related to the first foot problem was retained for analysis.

### Variables

The following baseline data were extracted from the patients' medical file by the treating physician: age, sex, diabetes type and duration, smoking status, relevant medical history, referral pattern, type of foot problem, ulcer location and severity of the index DFU according to the Perfusion, Extent, Depth, Infection and Sensation (PEDIS) classification system [15]. The information on the sex of the patient was recorded by the treating physician and classified as 'male', 'female' or 'unknown'. In case the patient was included in an earlier data collection, medical history and stable variables were validated against previous records in the database. During follow-up, information on index DFU management (offloading, vascular diagnostics, revascularization, orthopedic surgery and podiatric interventions) and outcomes (DFU healing, major amputation or death; relapse or new DFU) were recorded. Ulcer healing was defined as complete epithelialization with or without minor amputation (amputation below the ankle). Major amputation was defined as an amputation above the ankle, after which heel support is no longer possible. Full details of the questionnaire are described in the publicly available IQED-Foot report (Dutch/French) [16].

### Statistical analysis

Data analyses were performed using Statistical Analysis System (SAS) 9.4 (SAS Institute, Inc. Cary, NC). Differences in means and proportions between women and men were statistically tested with Generalized Estimating Equations, using the logit link function, an interchangeable correlation structure and robust standard errors (GENMOD procedure). This analysis took into account that responses were correlated within DFCs and resulted in appropriate inflation of standard errors, preventing overly optimistic conclusions compared to a standard generalized linear model approach. Non-parametric variables were transformed and statistical differences were assessed with the GENMOD procedure. Missing values were not taken into account, therefore the denominator reflects the number of registrations with a known value. Patients were considered lost to follow-up when the first and last contact date were the same and these patients were excluded from the outcome analyses. Results were expressed as a proportion, a mean (± standard error [SE]) for normally distributed variables or a median (25th percentile (P25)– 75th percentile (P75)) for non-normally distributed variables. Statistical significance was defined as $p < 0.05$.

A time-to-event analysis was performed for each ulcer-related outcome (DFU healing, major amputation or death) with the calculation of a hazard ratio (HR) taking into account the competing risk of the other two outcomes as a first event using a Cox proportional hazards regression model. The HR was adjusted for general characteristics (age, diabetes duration,

diabetes type and smoking status), comorbidities (renal insufficiency, kidney transplantation, cardiovascular disease, and revascularization), wound severity (Wagner grade and PEDIS), referral delay and follow-up time. HRs are reported with their 95% confidence interval (CI).

## Results

### Patient characteristics and referral pattern

The majority of the study population (72.0%) were male (Table 1). Women were significantly older than men. Smoking was more common in men than in women. These differences were most pronounced in the older age categories. More than 90% of women aged 75 years or older had never smoked, compared to 36.5% of men. This proportion decreased to 69% in women and 33.8% in men with an age between 65 and 74 years. In younger age categories, the proportion of smokers and never smokers was similar between the sexes. More men than women presented with a history of lower limb revascularization or a history of Charcot foot. Renal insufficiency was more common in women, while end-stage renal disease did not differ between sexes.

Patient referral and presentation delay were recorded for 1,230 (96.4%) and 1,225 (96.0%) men and 473 (95.6%) and 468 (94.5%) women, respectively. No differences in referral pattern were observed. The majority of individuals were referred by a health care professional, while 36.1% of men and 33.2% of women came to the DFC on their own initiative. The median [P25-P75] presentation delay, being the time between the self-reported onset of

**Table 1. Characteristics and medical history of patients at presentation.**

| | | All (n = 1,771) | Men (n = 1,276) | Women (n = 495) | p-value men vs women |
|---|---|---|---|---|---|
| Age, mean (SE) | | 69.7 (0.3) | 68.4 (0.3) | 73.0 (0.5) | <0.0001 |
| Diabetes type, n (%) | Type 1 | 136 (7.7) | 97 (7.6) | 39 (7.9) | 0.8444 |
| | Type 2 | 1,597 (90.2) | 1,148 (90.0) | 449 (90.7) | 0.5614 |
| | Other type | 38 (2.1) | 31 (2.4) | 7 (1.4) | 0.2035 |
| Diabetes duration, median (P25-P72) | | 16.4 (8.6–24.0) | 16.0 (8.3–23.0) | 17.6 (9.3–27.5) | 0.0017 |
| Smoking status, n (%) | Never | 756 (47.4) | 421 (36.6) | 335 (75.6) | <0.0001 |
| | Ex-smoker | 563 (35.3) | 505 (43.9) | 58 (13.1) | <0.0001 |
| | Smoker | 275 (17.3) | 225 (19.5) | 50 (11.3) | <0.0001 |
| Renal insufficiency*, n (%) | | 729 (42.2) | 503 (40.5) | 226 (46.8) | 0.0104 |
| End-stage renal disease[†], n (%) | | 170 (9.9) | 116 (9.3) | 54 (11.2) | 0.2495 |
| Cardiovascular disease[‡], n (%) | | 670 (41.3) | 496 (42.5) | 174 (38.1) | 0.2615 |
| Revascularization lower limbs, n (%) | | 575 (33.9) | 431 (35.2) | 144 (30.4) | 0.0249 |
| Previous ulcer, n (%) | | 918 (51.8) | 671 (52.6) | 247 (49.9) | 0.3515 |
| Previous Charcot, n (%) | | 116 (6.5) | 92 (7.2) | 24 (4.8) | 0.0446 |
| Previous minor amputation, n (%) | | 394 (22.2) | 300 (23.5) | 94 (19.0) | 0.0839 |
| Previous major amputation, n (%) | | 69 (3.9) | 56 (4.4) | 13 (2.6) | 0.0832 |

Proportions are expressed as percentages of known values.

[a] Defined as Modification of Diet in Renal Disease (MDRD) eGFR < 60 ml/min/1.73 m$^2$.

[b] Defined as renal transplantation or peritoneal or hemodialysis.

[c] Defined as history of myocardial infarction, coronary artery bypass grafting, percutaneous coronary intervention, stroke or transient ischemic attack.

DFU and the first contact in the DFC, was 3.0 weeks and did not differ between men and women (3.0 [1.0–7.7] vs 3.0 [1.0–8.0] weeks; p = 0.45). Patients with a history of DFU presented earlier compared to those with no history of DFU (2.6 [1.0–6.0] vs 3.9 [1.7–10.0] weeks) and more frequently presented on their own initiative (45.0% vs 24.1%). No differences were observed between sexes.

## Ulcer characteristics

Lesion burden was similar between sexes: 70.8% and 71.9% of men and women, respectively, had one lesion (p = 0.52), while 15.0% and 15.4% had an additional ipsilateral lesion (p = 0.84) and 14.2% and 12.7% had an additional contralateral lesion (p = 0.38). Fig 1 shows the location of the index DFU. In both men and women, most ulcers were located on the toes. However, men were significantly more likely to present with a plantar forefoot ulcer compared to women (26.8% vs 18.0%, p<0.0001). No differences were found in the proportion of ulcers located on the plantar midfoot, heel, dorsum or malleolus. Men were more likely to have a DFU spread over multiple locations than women (Table 2).

DFU severity was recorded through the PEDIS classification (Table 2). More than half of the study population presented with peripheral arterial disease, regardless of sex. Critical ischemia was significantly more common in men. Women were more likely to present with an ulcer < 1 cm$^2$. The ulcers in men were deeper, more frequently displaying probe to bone, and were more frequently deeply infected. Twice as many men presented with a

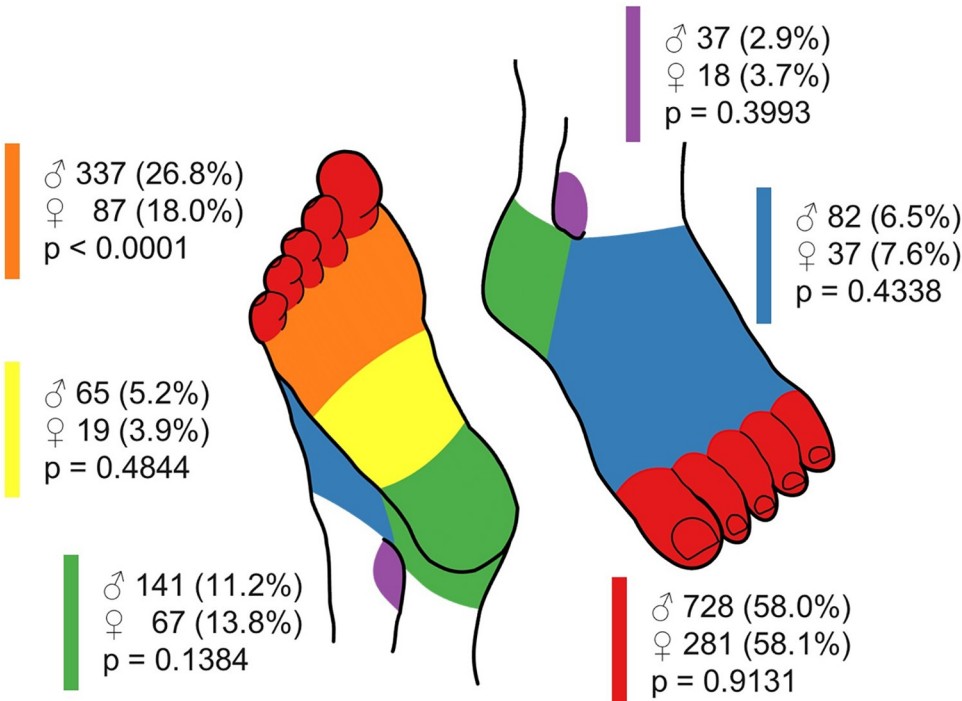

**Fig 1. Location of the index DFU.** A color-coded diagram was used by the DFC to indicate the location of the index DFU: Plantar forefoot (orange), plantar midfoot (yellow), heel (green), malleolus (purple), dorsum (blue) and toes (red). The number and proportion (%) of men and women with a DFU at a known location is indicated next to the respective color. The sum of percentages can exceed 100%, as a DFU can span multiple locations. The proportions of both groups were compared using generalized estimating equations.

**Table 2. Ulcer severity according to the PEDIS classification.**

| | | All (n = 1,771) | Men (n = 1,276) | Women (n = 495) | p-value men vs women |
|---|---|---|---|---|---|
| DFU side, n (%) | Right | 895 (50.5) | 656 (51.4) | 239 (48.3) | 0.1984 |
| DFU location, n (%) | > 1 location | 122 (7.0) | 101 (8.0) | 21 (4.3) | 0.0107 |
| Ulcer severity according to PEDIS | | | | | |
| Perfusion, n (%) | No PAD | 764 (45.3) | 553 (45.5) | 211 (44.8) | 0.9511 |
| | Subcritical ischemia | 683 (40.5) | 478 (39.3) | 205 (43.5) | 0.3252 |
| | Critical ischemia | 240 (14.2) | 185 (15.2) | 55 (11.7) | 0.0416 |
| Extent, n (%) | $< 1\ cm^2$ | 498 (29.2) | 337 (24.7) | 161 (34.2) | 0.0156 |
| | $\geq 1\ cm^2$ and $< 3\ cm^2$ | 797 (46.8) | 592 (48.1) | 205 (43.5) | 0.1290 |
| | $\geq 3\ cm^2$ | 408 (24.0) | 303 (24.6) | 105 (22.3) | 0.2781 |
| Depth, n (%) | Superficial | 235 (13.7) | 161 (13.0) | 74 (15.6) | 0.4955 |
| | Deep | 935 (54.6) | 660 (53.4) | 275 (57.9) | 0.0594 |
| | To bone | 542 (31.7) | 416 (33.6) | 126 (26.5) | 0.0033 |
| Infection, n (%) | No infection | 490 (28.5) | 335 (27.1) | 155 (32.2) | 0.0196 |
| | Superficial | 553 (32.2) | 385 (31.1) | 168 (34.9) | 0.3231 |
| | Deep | 586 (34.1) | 442 (35.7) | 144 (29.9) | 0.0441 |
| | Systemic | 89 (5.2) | 75 (6.1) | 14 (2.9) | 0.0100 |
| Sensation, n (%) | No protective sensation | 1,370 (85.9) | 1,008 (87.3) | 362 (82.5) | 0.0103 |

Proportions are expressed as percentages of known values.

DFU: Diabetic foot ulcer; PAD: Peripheral arterial disease.

systemic infection compared as women. Loss of protective sensation was more prevalent in men.

## Ulcer outcome

A total of 64 (3.6%) patients were lost to follow-up, of which 38 men and 26 women. All other patients not lost to follow-up were included in the outcome analysis. After a median (P25-P75) follow-up time of 154 (81–184) days, approximately half of the patients had healed, with or without minor amputation (Fig 2). Although not statistically significant, a slightly higher proportion of women had a healed DFU without any amputation at the end of follow-up compared to men (Fig 2A; 39.2% vs 34.6%, p = 0.05). 10.0% of men and 9.4% of women healed with a minor amputation (p = 0.45) (Fig 2B). Less than 5% of the study population underwent a major amputation (Fig 2C), with no differences between sexes (men 4.3% vs women 3.2%, p = 0.34). No sex differences were observed in the mortality rates (men 6.5% vs women 8.5%, p = 0.10) (Fig 2D). In addition, no differences were found in the median time to an event (healing, amputation or death) between sexes (Fig 2F–2I).

The Cox proportional regression analysis with competing risks demonstrated that women were more likely to heal without major amputation as a first event (crude HR: 1.156 (95% CI 0.999–1.338); adjusted HR: 1.258 (95% CI 1.048–1.509)). On the contrary, no significant association was found between sex and major amputation (crude HR: 0.748 (95% CI 0.394–1.420); adjusted HR: 1.117 (95% CI 0.472–2.640)) or death (crude HR: 1.071 (95% CI 0.710–1.615); adjusted HR: 1.054 (95% CI 0.537–2.069)) as a first event.

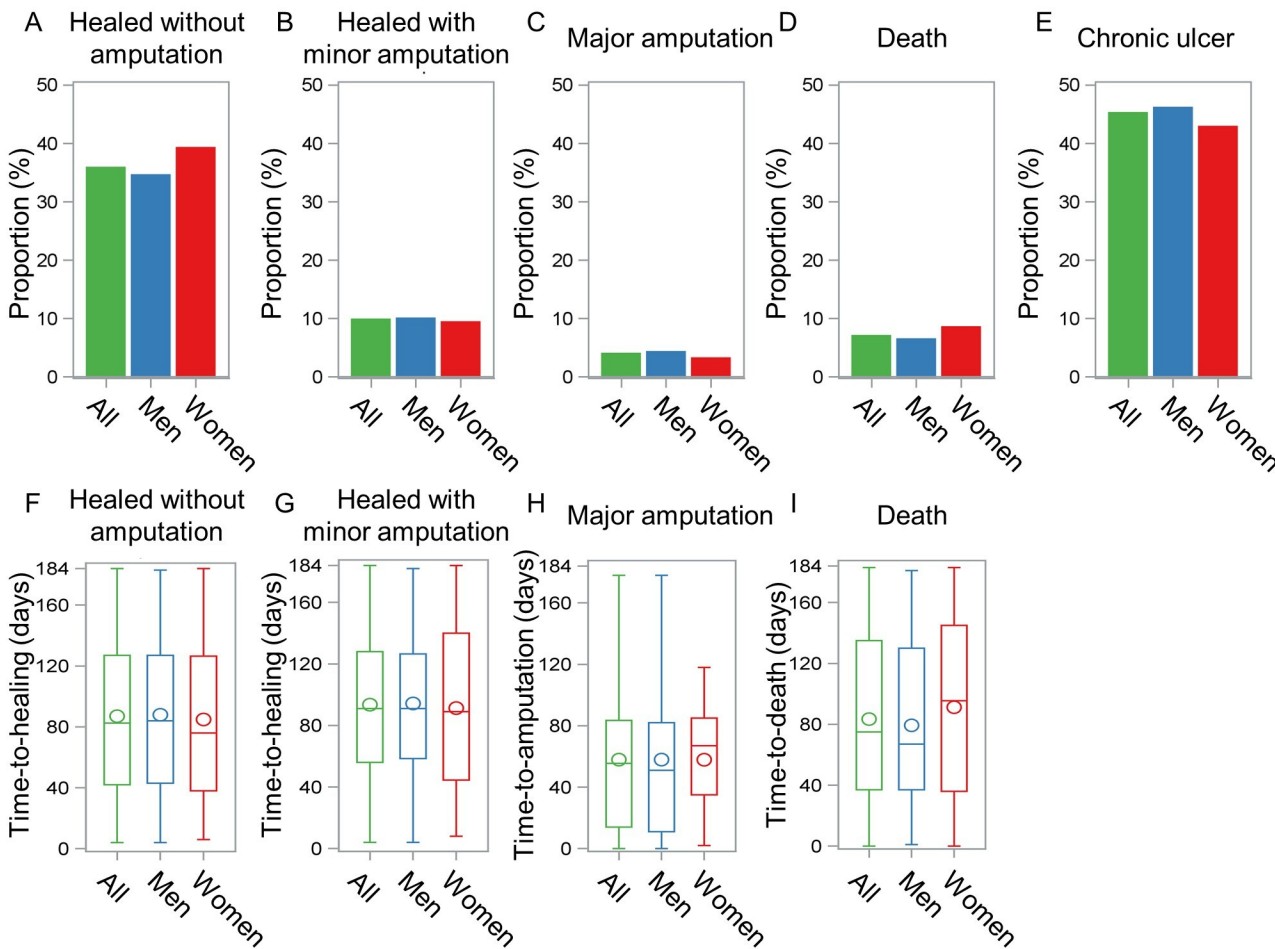

**Fig 2. Outcome of the index DFU after 6 months for patients not lost to follow-up.** A-E. Proportion of all patients (green, n = 1,707), male patients (blue, n = 1,238) or female patients (red, n = 469) who achieved the specified outcome during the follow-up period of maximum 6 months, being A. Healing without any amputation; B. Healing with minor amputation; C. Resolved by major amputation; D. Death or E. Chronic ulcer. Deaths were recorded throughout the follow-up period, bringing the total sum of the percentages above 100%, as some patients deceased after healing or major amputation. F-I. Boxplots showing the spread of the time (in days) to an event, being F. Healing without any amputation; G. Healing with minor amputation; H. Major amputation or I. Death. The median time-to-event is indicated by the horizontal line, the mean time-to-event by the circle.

## Discussion

Sex differences are increasingly recognized to play an important role in many aspects of health [7]. This study focused on sex-related differences in a cohort, followed in multidisciplinary diabetic foot clinics in Belgium.

### Patient characteristics

Almost three quarters of this study population with moderate to severe DFU was male. This appears to be in line with observations in the literature identifying male sex as a risk factor for the development of DFU [8]. This can be attributed, at least in part, to sex differences in disease awareness and self-care. It has been suggested that men are less likely to report chronic disease, indicating reduced disease awareness [17]. Women, on the other hand, are more attentive to symptoms and seek professional care sooner and more frequently than men [10,18]. In the context of DFU, women perform foot self-care more accurately [9–12], although men are less likely to use inappropriate footwear [10].

Women were on average 5 years older than men. However, long diabetes duration rather than age could be a risk factor for the development of foot complications [8]. Both men and women presented with longstanding diabetes. Although clinically not very relevant, a difference of 1.6 years in median diabetes duration was observed in the current study.

Substantial differences in the general smoking habits between men and women were found. These differences were most pronounced in the older age categories. The sex and age distribution of smoking habits in individuals over 65 years old in our study population is consistent with the Belgian national data. In contrast, the national data show a higher proportion of never smokers in the female population aged 45–54 years and 55–64 years compared to the current cohort (62.2% and 50.6% vs 44.4% and 43.9%, respectively). This could indicate a higher tobacco use among women aged 45–64 who presented with a DFU compared to the national population in 2018 in Belgium [19].

Men and women in this study demonstrated a differing pattern of co-morbidities that are key aspects of DFU.

Firstly, although both sexes presented a similar proportion of clinical symptoms of peripheral atrial disease (PAD), men more frequently presented with critical limb ischemia, indicating a higher prevalence of severe PAD in men in this cohort. The major risk factors for PAD are well known and include advanced age, diabetes, and tobacco use. Remarkably, sex could not be identified as a risk factor in a recent meta-analysis [20]. On the other hand, women are more likely to present with asymptomatic disease, resulting in later diagnosis in more advanced stages of PAD [21]. In addition, women are less likely to undergo lower limb revascularization [22], an element also reflected in the current study cohort. Unlike cardiovascular disease, there is no evidence that diabetes poses a higher excess risk for the development of PAD in women compared to men [23].

In general, smoking is associated with a worse cardiovascular state and a 25% greater increase of cardiovascular risk in women [24]. Furthermore, having diabetic foot problems in itself is a risk factor for cardiovascular complications, especially in women [25]. A higher prevalence of cardiovascular disease in women was not present in the current study, possibly due to the very high cardiovascular risk in both sexes. Smoking is considered a risk factor for worse outcomes in individuals with diabetic foot problems [8].

Secondly, a significantly higher proportion of women presented with renal insufficiency compared to men. This observation is in accordance with the literature. Men with diabetes, especially those with a diabetes duration of more than 25 years, appear to be at higher risk for diabetic nephropathy than premenopausal women. In contrast, postmenopausal women are at increased risk compared to men [26]. The vast majority ($> 94\%$) of women included in the current study were 55 years or older, therefore we can assume that they were indeed postmenopausal. In Belgium, national data on renal insufficiency are available for a subset of individuals with diabetes using 3 or more insulin injections per day or pump therapy. Note that due to the sampling conditions of these national data, the proportion of people with type 1 diabetes mellitus (30%) is higher compared to our study cohort (8%). In this national data set, women also present with renal insufficiency more frequently than men (41.0%, mean age 66.2 years vs 33.5%, mean age 64.1 years, p<0.0001) [27]. A higher prevalence of end-stage renal disease has been reported in women than men [26]. However, this was not the case in our study cohort, nor in the national data set [27]. The exact mechanisms underlying the preponderance of female sex in the development and progression of diabetic nephropathy are not yet fully understood. The proposed mechanisms include differences in sex hormones, hemodynamics of the kidney, adiponectin concentrations and concomitant risk factors, such as smoking [26].

## Ulcer characteristics

In this study, men presented with more severe DFU compared to women. Ulcers were larger, deeper and significantly more frequently associated with osteomyelitis or systemic infection. Although osteomyelitis and male sex have been identified as individual risk factors for amputation [28,29], patient sex does not affect the likelihood of DFU infection [30]. Some studies indicate that men are more prone to developing surgical site infections [31]. At a biological level, there are sex-related differences in the immune response to infection, as highlighted by the Coronavirus Disease 2019 (COVID-19) pandemic [32]. Another explanation may be the sex-related differences in health-related behaviors, which may lead to differences in the timeliness in which care is sought for acute problems such as infection [17]. Remarkably, in our study population, the most severe DFU was not associated with a longer presentation delay in men, nor with the proportion of individuals referred by a healthcare professional. It should, however, be noted that presentation delay was a patient-reported variable and may be susceptible to recall bias.

Approximately half of men and women had a history of previous DFU. Recurrence rates are indeed known to be very high, with an estimated 60% within 3 years, and 65% within 5 years [2]. A recent meta-analysis demonstrated that the male sex is associated with an increased incidence of DFU recurrence [33]. Nonetheless, it should be noted that in many studies on recurrence, sex is not taken into consideration in the statistical analysis [34,35]. Remarkably, a history of Charcot foot was significantly higher in men. The data in the literature on sex-related differences in the prevalence of Charcot foot remain controversial [36].

A significantly higher proportion of plantar forefoot DFU was seen in men. Obese individuals have been previously shown to have increased plantar peak pressures, with the highest effects in the plantar forefoot and midfoot regions [37,38]. However, no data on weight or body mass index were collected in the current study. The overall effect of sex on plantar pressure is not clear, as one study suggested that female sex is associated with changes in peak pressure in the hindfoot and forefoot region [39], while others reported an association with abnormal pressure distribution at the lateral part and midfoot [38]. Elevated forefoot pressure may also result from diabetic neuropathy [39]. In our study cohort, men indeed presented slightly more frequently with loss of protective sensation compared to women. This observation is consistent with data from the literature, indicating that diabetic neuropathy is more common and develops earlier in men than in women [40].

## Ulcer outcome

Our analysis demonstrated, although not significant, a slightly higher healing rate without any amputation after 6 months of follow-up for women compared to men. After adjusting for ulcer severity and patient characteristics, female sex was significantly associated with a higher probability of ulcer healing as a first event. Minor and major amputation rates did not differ between the sexes. The latter observation is in contrast with previous studies in which men with diabetes are at higher risk of undergoing amputations, both minor and major [28,29]. In the literature, this higher risk is attributed to the fact that men are more likely to have risk factors for lower limb amputation, such as tobacco use, PAD, peripheral neuropathy, deep and infected DFU [28]. Interestingly, socio-economic status also appears to have a greater impact on amputation risk in men than in women [41].

Several studies suggest that female and male patients are treated differently to some extent. Women with diabetes are less likely to reach targets for cardiovascular prevention, such as lipid, blood pressure and glycated hemoglobin (HbA1c), less likely to undergo lower limb revascularization, less likely to be monitored for foot and eye complications, and less likely to

be reminded to wear their therapeutic shoes by clinical staff [9,22,40,42]. This is even further complicated by the observation that some pharmacological treatments have different efficacy between sexes [40,43,44]. Moreover, the literature also shows that the sex of the treating physician can influence the outcome, a factor that puts particularly female patients at risk when they are treated by a male physician [45,46].

These observations indicate that health care professionals should be aware of the effect of sex differences in the prevention, treatment and follow-up of DFU. Moreover, the impact of sex on the patient's own perception of their disease and the care they receive should not be neglected. These differences should be taken into account to optimize preventive and therapeutic strategies for diabetes and diabetic foot care in a more sex-specific way.

## Strengths and limitations

An important strength of the study is the nationwide data collection with a large number of "real-world" observations. Moreover, the observational data are collected in a prospective manner, hereby reducing the risk of bias. However, we also acknowledge some limitations. First, the study only included information on moderate to severe DFU, which could have resulted in an overestimation of comorbidities and DFU severity. A second limitation is that, although many parameters were recorded in the IQED-Foot database, no data were collected on body mass index, glycated hemoglobin, lipid status, socio-economic status, lifestyle profile or patient-reported experience measures, all of which are of interest when studying a sex-related effect.

## Conclusion

To conclude, in our study population, men presented with more severe DFU than women, although no increase in presentation delay was observed. Moreover, female sex was significantly associated with a higher probability of ulcer healing as a first event. Of the many possible contributing factors, a worse vascular state associated with a higher rate of (previous) smoking in men stands out. These findings suggest that attention to sex should be included in both research and clinical optimization of preventive and therapeutic strategies in the treatment of diabetic foot ulcers.

## Acknowledgments

The authors would like to thank the diabetic foot clinics for their participation in the IQED-Foot data collection.

Members of the IQED-Foot Study Group: An-Sofie Vanherwegen (lead author, E-mail: iqedfoot@sciensano.be), Flora Mbela Lusendi, Sciensano, Brussels; Dimitri Aerden, Nathalie Denecker, University Hospital Brussels, Brussels; Sabine De Bruyne, Cédric Coucke, AZ ST-Lucas, Gent; Jean-Philippe De Wilde, Christophe Jacobs, Hôpital Erasme, Anderlecht; Kevin Deschamps, Sabrina Houthoofd, and Giovanni Matricali, University Hospital Leuven, Leuven; Sophie Deweer, Boudewijn Moors, Sint-Elisabethziekenhuis, Zottegem; Eveline Dirinck, Patrick Lauwers, University Hospital Antwerp, Antwerp; Isabelle Dumont, Centre du Pied de Ransart, Ransart; Vincent Ers, Eric Weber, Cliniques du Sud Luxembourg, Arlon; Patricia Félix, CHR de la Citadelle, Liège; Olga Kosmopoulou, Antoine Pigeon, CHU Brugmann, Laken; Philippe Lerut, Gertjan Vereecke, AZ Groeninge, Kortrijk; Hilde Beele, Cédric Lannoo, Nathalie Moreels, Caren Randon, Steven Smet, University Hospital Gent, Gent; Frank Nobels, Onze-Lieve-Vrouwziekenhuis, Aalst; Marcelle Rorive, CHU du Sart Tilman, Liège; Viviane Van Elshocht, Landsbond der Christelijke mutualiteiten, Schaarbeek; Michel Vandenbroucke, AZ Sint-Maarten, Mechelen.

## Author Contributions

**Conceptualization:** An-Sofie Vanherwegen, Eveline Dirinck.

**Data curation:** An-Sofie Vanherwegen, Astrid Lavens.

**Formal analysis:** An-Sofie Vanherwegen, Astrid Lavens.

**Funding acquisition:** An-Sofie Vanherwegen.

**Methodology:** An-Sofie Vanherwegen.

**Supervision:** Eveline Dirinck.

**Visualization:** An-Sofie Vanherwegen.

**Writing – original draft:** An-Sofie Vanherwegen.

**Writing – review & editing:** An-Sofie Vanherwegen, Patrick Lauwers, Astrid Lavens, Kris Doggen, Eveline Dirinck.

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
