## [Decision Letter · Decision Letter 0]

14 Dec 2022

PONE-D-22-31378Sex differences in diabetic foot ulcer severity and outcome in BelgiumPLOS ONE

Dear Dr. Vanherwegen,

Thank you for submitting your manuscript to PLOS ONE. After careful consideration, we feel that it has merit but does not fully meet PLOS ONE’s publication criteria as it currently stands. Therefore, we invite you to submit a revised version of the manuscript that addresses the points raised during the review process.

We look forward to receiving your revised manuscript.

Kind regards,

Tariq Jamal Siddiqi

Academic Editor

PLOS ONE

Journal Requirements:

4. One of the noted authors is a group or consortium: Initiative for Quality Improvement and Epidemiology in Diabetic Foot Clinics Study Group

In addition to naming the author group, please list the individual authors and affiliations within this group in the acknowledgments section of your manuscript. Please also indicate clearly a lead author for this group along with a contact email address.

Reviewers' comments:

Reviewer's Responses to Questions

**Comments to the Author**

1. Is the manuscript technically sound, and do the data support the conclusions?

Reviewer #1: Yes

2. Has the statistical analysis been performed appropriately and rigorously? 

Reviewer #1: Yes

3. Have the authors made all data underlying the findings in their manuscript fully available?

Reviewer #1: Yes

4. Is the manuscript presented in an intelligible fashion and written in standard English?

Reviewer #1: Yes

5. Review Comments to the Author

Reviewer #1: The authors conducted a study to assess sex differences in diabetic foot ulcer severity and outcome in Belgium. The manuscript is drafted decently and data is presented in an intelligible manner. However, the following points must be addressed to improve the article further:

1. Please replace references that are more than five years old.

2. Clearly highlight the gaps in literature in the Introduction.

3. Specify the type of diabetes mellitus.

4. In lines 131 –132, cite the reference instead of inserting the link.

5. Lines 117-118: Please rephrase as “The following baseline data extracted from the medical file of the patient by the treating physician”

6. Use full forms of all abbreviations the first time they are used for example, PEDIS, SAS 9.4

7. Line 155: It should be “for a 95% confidence interval”

8. Please make the “Patient characteristics and referral pattern“ section of Results more concise, highlighting only the major findings since the rest is detailed in Table 1.

9. Line 190: “The majority of the ulcers was 191 located on the toes in both men and women. Men presented significantly more often 192 with a plantar forefoot ulcer compared to women (26.8% vs 18.0%, p<0.0001).” should be “In both men and women, the majority of ulcers were located on the toes with men presented significantly more often with a plantar forefoot ulcer compared to women (26.8% vs 18.0%, p<0.0001).”

10. Please remove symbols for male and female from Line 200.

11. For the “Ulcer outcome”, mention outcomes of patients not lost to follow up first and then for those not lost to follow-up. No independent headings needed for each, just a main heading of “Ulcer outcome” is fine. In this manner, results for patients not lost to follow-up can simply be stated as being similar between sexes for all the above-mentioned outcomes without mentioning each outcome (and its numbers) independently.

12. Line 230: “a” should be capitalized.

13. Mention the significant results obtained without regression as well.

14. Use better vocabulary where possible such as “demonstrated” or “depicted” instead of “showed” and “more frequently” instead of “more often”.

15. Line 243: It should be “.” instead of “,” in 1054.

16. While the manuscript is easy to read and understand, its language and vocabulary need to be enhanced.

17. Please improve the phrasing of the Discussion and make it more concise, partciularly while discussing patient characteristics.

18. Line 401: it should be “we also acknowledge”

6. PLOS authors have the option to publish the peer review history of their article (what does this mean?). If published, this will include your full peer review and any attached files.

Reviewer #1: No

---

## [Author Response · Author response to Decision Letter 0]

20 Jan 2023

Response to academic editor

Authors: The manuscript has been adapted in order to meet PLOS ONE’s style requirements described in the templates.

Authors: The IQED-Foot quality initiative is funded by the National Institute of Health and Disability Insurance through a convention agreement. There is no grant number appointed.

Due to the fact that Belgium has different official languages, the National Institute of Health and Disability Insurance is also named Rijksinstituut voor Ziekte- en Invaliditeitsverzekering (Dutch) or Institut National d'Assurance Maladie-Invalidité (French), which might have caused the mismatch between the ‘Funding Information’ and ‘Financial Disclosure’ sections.

The funding information section has been updated to:

The IQED-Foot quality initiative is funded by the National Institute of Health and Disability Insurance. The funders had no role in study design, data collection and analysis, decision to publish, or preparation of the manuscript.

The IQED-Foot database contains sensitive, personal data. The access, storage and sharing of the data is strictly regulated by the Belgian Information Security Committee Social Security and Health. Data can only be shared in an anonymous way. However, de-identifying the used dataset for the current manuscript does not guarantee complete absence of re-identification of the patients based on several indirect identifiers used for the analyses. For this reason, we are not able to make the de-identified dataset publically available. We propose to update the data availability statement as follows:

Data availability statement

Data cannot be shared publicly because of the use of pseudonomysed patient data. Actors wanting to access (parts of the) data require an approval from the Belgian Information Security Committee Social Security and Health. For more information about the access procedure: iqedfoot@sciensano.be. Metadata (e.g. overview of variables, legal framework) are available on https://fair.healthdata.be/

4. One of the noted authors is a group or consortium: Initiative for Quality Improvement and Epidemiology in Diabetic Foot Clinics Study Group

In addition to naming the author group, please list the individual authors and affiliations within this group in the acknowledgments section of your manuscript. 

Please also indicate clearly a lead author for this group along with a contact email address.

Authors: The members of the Initiative for Quality Improvement and Epidemiology in multidisciplinary Diabetic Foot Clinics (IQED-Foot) Study Group and their affiliations are mentioned in the acknowledgement section of the paper. A lead author was indicated along with a contact mail address (An-Sofie Vanherwegen, Sciensano; iqedfoot@sciensano.be).

 

Response to Reviewers

Reviewer #1: The authors conducted a study to assess sex differences in diabetic foot ulcer severity and outcome in Belgium. The manuscript is drafted decently and data is presented in an intelligible manner. 

Authors: We would like to thank reviewer #1 for his/her time to evaluate our manuscript in such great detail.

However, the following points must be addressed to improve the article further:

1. Please replace references that are more than five years old. 

Authors: We acknowledge that it is favorable to build further on the most recent literature available. However, as the literature on gender differences in diabetic foot care is rather limited, it is not always possible to refer to publications within the past 5 years to support a specific observation. In addition, we do see an added value to refer to the original research article rather than a recent review that would not contain additional relevant information except for the reference to the original article. In this regard, we only replaced the following selection of references that are more than 5 years old: 

- Monteiro-Soares et al. 2012 � Rodrigues et al. 2022

- Teodorescu et al. 2013 � Srivaratharajah et al. 2018

- Huxley et al. 2011 � Humphries et al. 2017

- Pickwell et al. 2015 � Lin et al. 2020

- Dubsky et al. 2013 � Huang et al. 2019

- Fauzi et al. 2016, Younis et al. 2015, Sohn et al. 2009, Nehring et al. 2014 � Zhao et al. 2017

- Hills et al. 2001 � Sutkowska et al. 2019

- Tang et al. 2014 � Lin et al. 2020

2. Clearly highlight the gaps in literature in the Introduction. 

Authors: The following text was added in the introduction to highlight the gaps in literature (lines 77-80):

Despite the acknowledgement of a negative role of male sex in the onset of DFU [8], the literature is less extensive on the differences in clinical presentation and outcomes between men and women. However, a better understanding will contribute to the optimization of care for this diabetes complication in a sex-specific manner.

3. Specify the type of diabetes mellitus.

Authors: Any type of diabetes mellitus could have been included in the study population. Upon inclusion, the clinician registered the type of diabetes mellitus as type 1, type 2 or other type. 

In our study population, 7.7% of the people had type 1 diabetes, 90.2% type 2 diabetes and 2.1% another type of diabetes (data shown in Table 1).

The text in line 110 was adapted as follows:

To be included in the study population, individuals had to be 18 years old or older, have diabetes mellitus (type 1, type 2 or other) and present…

4. In lines 131 –132, cite the reference instead of inserting the link. 

Authors: The link was replaced by a reference (line 132).

5. Lines 117-118: Please rephrase as “The following baseline data extracted from the medical file of the patient by the treating physician”

Authors: The text in lines 118-119 was adapted as suggested.

6. Use full forms of all abbreviations the first time they are used for example, PEDIS, SAS 9.4

Authors: We have revised the text and added the full form where missing.

Line 106: General Data Protection Regulation (GDPR)

Line 123: Perfusion, Extent, Depth, Infection and Sensation (PEDIS)

Line 137: Statistical Analysis System (SAS) 

Line 149: 25th percentile (P25)

Line 150: 75th percentile (P75)

Line 320: Coronavirus Disease 2019 (COVID-19)

Line 364: glycated hemoglobin (HbA1c)

7. Line 155: It should be “for a 95% confidence interval” 

Authors: Diabetes duration, referral delay and follow-up time are not normally distributed and therefore expressed as a median with 25th and 75th percentile. We noticed that this was not correctly written for referral delay in the second paragraph of the ‘patient characteristics and referral pattern’ section (lines 180-183).

The 95% confidence interval is only being reported for the Hazard Ratio.

The text in lines 180-183 was adapted as follows:

The median [P25-P75] presentation delay, being the time between the self-reported onset of DFU and the first contact in the DFC, was 3.0 weeks and did not differ between men and women (3.0 [1.0–7.7] vs 3.0 [1.0–8.0] weeks; p=0.45). Patients with a history of DFU presented earlier compared to those with no history of DFU (2.6 [1.0–6.0] vs 3.9 [1.7–10.0] weeks)…

The text in lines 146-149 was adapted as follows:

Results were expressed as a proportion, a mean (± standard error [SE]) for normally distributed variables or a median (25th percentile (P25) – 75th percentile (P75)) for non-normally distributed variables. Statistical significance was defined as p < 0.05.

The text in line 157 was adapted as follows:

HRs are reported with their 95% confidence interval (CI).

8. Please make the “Patient characteristics and referral pattern“ section of Results more concise, highlighting only the major findings since the rest is detailed in Table 1. 

Authors: The first paragraph of the ‘patient characteristics and referral pattern’ section was made more concise as requested.

9. Line 190: “The majority of the ulcers was 191 located on the toes in both men and women. Men presented significantly more often 192 with a plantar forefoot ulcer compared to women (26.8% vs 18.0%, p<0.0001).” should be “In both men and women, the majority of ulcers were located on the toes with men presented significantly more often with a plantar forefoot ulcer compared to women (26.8% vs 18.0%, p<0.0001).” 

Authors: The text in lines 190-192 was adapted as suggested.

10. Please remove symbols for male and female from Line 200. 

Authors: The symbols in the legend of Figure 1 were removed as requested.

11. For the “Ulcer outcome”, mention outcomes of patients not lost to follow up first and then for those not lost to follow-up. No independent headings needed for each, just a main heading of “Ulcer outcome” is fine. In this manner, results for patients not lost to follow-up can simply be stated as being similar between sexes for all the above-mentioned outcomes without mentioning each outcome (and its numbers) independently. 

Authors: Unfortunately, due to the prospective design of the data collection, we are not able to report the outcomes of patients that are lost to follow-up as this means that the patient only had one consultation at the diabetic foot clinic at the time of presentation. No outcomes have been registered for those patients and they were therefore removed from the denominator of the outcome analyses.

This is mentioned as follows in the Methods section (lines 144-146):

Patients were considered lost to follow-up when the date of first and last contact date were the same and these patients were excluded from the outcome analyses.

We have added the following sentence in line 218 of the ‘Ulcer outcome’ paragraph to make the exclusion of patients lost to follow-up clearer: 

All other patients not lost to follow-up were included in the outcome analysis.

12. Line 230: “a” should be capitalized. 

Authors: The text in line 233 was adapted as suggested.

13. Mention the significant results obtained without regression as well. 

The statistical comparison between men and women for the DFU outcomes (Figure 2) is mentioned in the section ‘Ulcer outcome’ lines 220 – 226. Only a trend (p=0.05) towards higher healing rates without amputation could be observed in women (39.2%) compared to men (34.6%). The crude hazard ratios from the Cox proportional regression analysis equaled to 1.156 (95% CI 0.999-1.338) for healing without major amputation, 0.748 (95% CI 0.394-1.420) for major amputation and 1.071 (95% CI 0.710-1.615) for death as first event.

We have added the crude HRs in the text lines 243-247.

14. Use better vocabulary where possible such as “demonstrated” or “depicted” instead of “showed” and “more frequently” instead of “more often”. 

Authors: We have revised the text and made changes where needed.

15. Line 243: It should be “.” instead of “,” in 1054. 

Authors: The format of the numbers in line 247 and Table 1 was adapted.

16. While the manuscript is easy to read and understand, its language and vocabulary need to be enhanced. 

Authors: The manuscript was revised by an external proofreader (proof of the external proofreading included in the file "Response to reviewers"). We hope that the quality of the language and vocabulary enhanced sufficiently.

17. Please improve the phrasing of the Discussion and make it more concise, partciularly while discussing patient characteristics. 

Authors: The Discussion section was made more concise as requested. The manuscript was revised by an external proofreader. We hope that the phrasing enhanced sufficiently.

18. Line 401: it should be “we also acknowledge” 

Authors: The text in line 389 was adapted as suggested.

---

## [Editor Report · Decision Letter 1]

2 Feb 2023

Sex differences in diabetic foot ulcer severity and outcome in Belgium

PONE-D-22-31378R1

Dear Dr. Vanherwegen,

We’re pleased to inform you that your manuscript has been judged scientifically suitable for publication and will be formally accepted for publication once it meets all outstanding technical requirements.

Kind regards,

Tariq Jamal Siddiqi

Academic Editor

PLOS ONE
---

## [Editor Report · Acceptance letter]

7 Feb 2023

PONE-D-22-31378R1 

Sex differences in diabetic foot ulcer severity and outcome in Belgium 

Dear Dr. Vanherwegen:

I'm pleased to inform you that your manuscript has been deemed suitable for publication in PLOS ONE. Congratulations! Your manuscript is now with our production department. 

Kind regards, 

on behalf of

Dr. Tariq Jamal Siddiqi 

Academic Editor

PLOS ONE